



# A New Approach for Rainfall Rate Field Space-Time Interpolation for Western Europe

Guangguang Yang[1], David Ndzi[2], Kevin Paulson[3], Misha Filip[1]

[1]School of Engineering, University of Portsmouth, Portsmouth, PO1 3DJ, United Kingdom

[2]School of Engineering, University of West of Scotland, Paisley, United Kingdom

[3]School of Engineering, University of Hull, Hull, HU6 7RX, United Kingdom

guangguang.yang@myport.ac.uk; david.ndzi@uws.ac.uk; K.Paulson@hull.ac.uk; misha.filip@port.ac.uk

**Abstract.** The prediction of rainfall rate characteristics at small space-time scales is currently an important topic, particularly within the context of the planning and design of satellite network systems. A new comprehensive interpolation approach is presented in this paper to deal with such an issue. There are three novelties in the proposed approach: 1) the proposed interpolation approach is not directly applied to measured rain precipitation (either radar or raingauge-derived data) but focuses on the coefficients of the fitted statistical distributions and/or computed rain characteristics at each location; 2) the parameter databases are provided and the contour maps of coefficients spanning Western Europe have been created. It conveniently and efficiently provides the rain parameter for any location within the studied map; 3) more speculatively, the 3D space-time interpolation approach can extrapolate to rain parameters at space-time resolutions shorter than those in the NIMROD databases.

## 1. Introduction

The spatial and temporal variation of point rainfall rates is important for the detailed planning and performance prediction for satellite and terrestrial networks (a group of links) (Yang, 2016). It is increasingly evident that models and/or approaches are needed in order to predict rainfall rate variation at smaller space-time scales than currently available from wide area coverage measured rainfall rate databases.

Extensive studies of rain have been carried out in the last few decades. After several generations many interesting rain models have been published. A model of particular interest was developed by Bell (Bell, 1987). His work showed that rainfall intensities in a field exhibit lognormal distribution and this was confirmed by Crane (Crane, 1996) and Jeannin et al (Jeannin et al., 2008). The traditional rain models (e.g. stochastic models, Markov chain models) can be used to aid the planning of satellite networks. However, there are some limitations inherent in such models and the two major ones are:

1) Data availability. The models are only applicable to areas/locations where rainfall precipitation with the necessary integration volume has been observed and the accuracy of the models in areas where no data is available is difficult to verify.

2) Integration volume. The application of the traditional models is limited by the integration length. The modelling of rain and simulated rainfall fields can only be limited to the space-time resolution derived from rain radar/gauge measurements. Rainfall fields simulation at finer space-time scales is often possible but cannot be verified.

Based on this information, it is clear that the application range of stochastic models is limited by the above problems. Improvements, thus, are needed to compensate, enhance and extend the performance of stochastic models. In particular, an increase in the use of high frequency over short communication links has led to an increase in the need to predict rainfall rates at finer resolutions. Current stochastic models cannot satisfy this demand. As a result, interpolation techniques have attracted a lot of attention in recent decades. For example, Drozdov and Sephelevskii (Drozdov and Shepelevskii, 1946) developed a spatial interpolation technique to analyze the spatial variations of a process over an area. Then later, a modified interpolation technique



called Kriging was developed based on the theory of regionalized variables to estimate area averages considered as realizations

of a stochastic process introduced by Matheron (Matheron, 1971) Since then significant progress has been made and two-

dimensional (2D) space rainfall rate interpolation models have been developed, e.g.(Deidda,1999 and Menabde et al., 1997). The

Random Midpoint Displacement algorithm (RMD) developed by Voss (Voss, 1985) in 1985 is one of the most popular

interpolation algorithms. The basic idea of the technique is to introduce new rain rate samples with the same underlying

distribution as existing measurements at new locations or times. The one-dimensional (1D) time interpolation is also of interest

as network planners and designers of physical layer fade mitigation techniques (Gremont et al., 1999) require knowledge of rain

variation over much shorter time scales (of the order of seconds or less). Some excellent models have been published like

(Pathirana et al., 2003 and Veneziano et al., 1996). One of such models proposed by Kevin Paulson (Paulson, 2004) is a

stochastic numerical model that can interpolate the point rain rate for short time durations down to $10\ s$.

The downscaling model, based on the space-time averaging theory, is another model that has also attracted significant attention.

According to (Deidda et al., 1999), there are two fundamental requirements for precipitation downscaling models, which are: 1)

understanding of the statistical properties and scaling laws of rainfall fields, and 2) validation of downscaling models that are

able to preserve statistical characteristics observed in real precipitation. Typically, based on the information given in (Rebora et

al., 2006), downscaling algorithms can generally be grouped into three main families with some simplification: 1) point process

based on the random positioning of a given number of rain bands and rain cells (Cowpertwait et al., 2006); 2) autoregressive

processes passed through a static nonlinear transformation (Guillot and Lebel, 1999), and; 3) fractal cascades (Kiely and

Ivanova, 1999). In particular, the theory of fractals, which was first introduced by Mandelbrot in 1967 (Mandelbrot, 1967) has

attracted great attention. This theory was not applied to rainfall study until the mid-1980s (Lovejoy and Mandelbrot, 1985). Rain

has been shown to hold fractal properties over a range of scales. The intermittence and discontinuous nature of rain is reproduced

by the fractal based models, which are strongly favoured for rainfall modelling. Many studies have been carried out to interpolate

the radar/raingauge measurement data to finer scales using the fractal theory, such as (Svensson, 1996). Multifractal models,

which may be simulated using random cascades, can easily capture any moment of the observed signal; especially higher order

moments have attracted a lot of attention. Because of their link with multifractal theory, multiplicative cascade models first

proposed by Yaglom (Yaglom, Jul 1966), appeal to rainfall simuláations. The rainfall series have been shown to exhibit scaling

invariance properties over a large range of space (Olsson, 1996) and time (Olsson et al., 1993) steps. Some multifractal models

use discrete cascade algorithms to produce data at finer scales from original sparse observations, for example (Olsson, 1998). A

classic work is given by Menabde (Menabde, 1997) who used a discrete random cascade to generate a rain field with the desired

statistical structure and then applied a power law filter, thereby removing some of the blockiness resulting in a more realistic

looking rain field. In addition, synthesis of rain field at high resolution is also important to the rain study especially devised for

applications related to EM wave propagation. Many contributions have been done in this area, such as (Jeannin, 2012 and Luini,

2011)

The prediction at finer space-time resolution however, has long been a challenging issue in rainfall field modeling. Results from

3D interpolation studies are quite poor (Yang, 2016 and Deidda, 2000) as it is very difficult to consider both space and time

variability and irregularity of rainfall in an appropriate way. The basic idea of published models is to try to find the underlying

principle of how the space-time transformation can be achieved. A representative model was developed by Deidda (Deidda,

2000) based on the assumption that Taylor's hypothesis (Taylor, 1938) can be applied. The space-time rainfall field is assumed

to be a three-dimensional (2D space and 1D time) homogeneous and isotropic process. An advection velocity parameter is

introduced to connect the space scale and time scale. With the help of a velocity parameter, the statistical properties of rain at

finer scales can be deduced from larger ones. Similar studies can be found in (Venugopal et al, 1999, Deidda, 2006 and





Venugopal et al., 1999) in which rain has been studied in a range of space-time scales to define the transformation parameter. In
particular, Kundu and Bell (Kundu and Bell, 2006) developed a model that can provide the correlation function of rain in 3$D$
space-time domain but in a very complicated form.
The absence of high resolution rainfall data at desired space and time scales is the main knowledge gap. Deidda in (Deidda,
2000) pointed out that most of the existing rainfall studies at finer scales are purely focused on either space modeling (Hubert et
al., 1993) or time modeling (Paulson, 2004). However, both of these approaches have limitations. For example, the statistical
behavior of rain in time has implicit consideration of the spatial distribution and extension of the rain field itself; and the study in
space is normally based on fixed time duration whilst the evolution in time of spatial patterns is ignored. Accurate rainfall field
simulation requires knowledge of rainfall rate variability in both space and time domains. There is not enough research in the
area of space-time interpolation apart from a few works, such as (Deidda, 2006). Thus, an appropriate space-time interpolation
model that can preserve the underlying statistical properties at finer scales is needed. The absence of knowledge of rain
characteristics at high space and time resolution is another important gap and is the second objective of this study. Kundu in
(Kundu and Bell, 2006) showed that the characteristics of rain depend on the space and time scales over which rain data is
averaged. However, all the existing interpolation and/or multifractal models directly focus on rain precipitation and no work has
been found that studied the characteristics of rain at scales better than the one provided by rain radars. The study in this paper,
therefore, will look into this issue to investigate the variability of rain characteristics at arbitrary space-time integration length.
To further the development of rain-induced radio-wave attenuation models, and to provide more accurate performance prediction
of satellite links over wide areas, there is an increasing need for a good understanding of the space-time characteristics of rainfall
rate at finer scales. As extension of our previous work (Yang, 2011), this paper presents a simple but accurate space-time
interpolation approach that can interpolate the key studied properties of rain in both space and time domain simultaneously. We
present a series of European maps superimposed with each parameter at different space-time resolution which is novel. In
particular, a simple but accurate approach for interpolating the rain characteristics has been proposed. It can predict the
coefficient values of the statistical model in both space and time with reasonable accuracy.
The rest of this paper is organized as follows: Section 2 describes the data used in this study. Section 3 reviews the statistical
model proposed in previous work and describes the proposed approach how to interpolate the measurements into 3-dimensional
space-time domain. The detailed results, including the 2D contour map of rain characteristics across Western Europe, as well as
the 3D space-time predictions at each location, are presented in Section 4. Section 5 validates the results achieved from the
proposed interpolation approach. Conclusions are drawn in Section 6.
**2.   Data Description**
Five complete years of NIMROD rain radar data (from 2005 to 2009) have been analyzed for the development of a generic
interpolation approach. The NIMROD radar system produces a series of composite rain field map by every 15 $mins$. The
measured rain rate samples are distributed on a 5 $km$ squared Cartesian grid covering Western Europe. Each NIMROD map
contains 700 × 620 data cells, but only the data available points have been analysed, see the outline is Fig. 1(a). The study area
ranges from 43.1938° to 59.4306° in latitude and −9.7370° to 19.8364° in longitude. In addition, NIMROD system also holds
the database for the British Isles. This database has better resolution of rain rate measurement, which is 1 $km$ in space and
5 $mins$ in time. The example radar map is given in Fig. 1(b). The performance of any model or approach needs to be validated
through comparing with observational data from apparatus (e.g. raingauge or rain radar). UK data, which has better resolution
than EU NIMROD data, can be utilized to implement the validation.





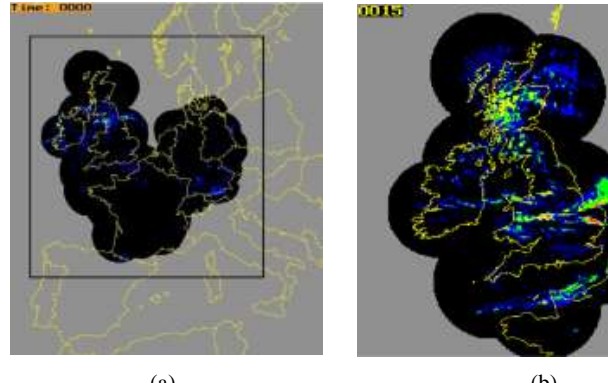

(a)                (b)

**Figure 1**: (a) composite radar scan image: radar image for Western Europe (the outline is the studied area), and (b) radar image
for the British Isles.
**3. Methodology**
**3.1. Stochastic Model**
The empirical equations that can accurately provide the estimates of the studied characteristics of rain have been discussed in 0.
The proposed model for the four key rain characteristics is described briefly here for completeness.
It is well accepted that rainfall rate $R$ in mm/h at one location is modeled as a lognormal process with mixed probability density
function (pdf). According to (Filip and Vilar, 1990), the general formula for a straight line fit is given by:
$Q_{inv} = \frac{ln(R)}{\sigma} + \frac{\mu}{\sigma}$          (1)
where $\{\mu, \sigma\}$ is the set of lognormal parameters that are used to study the statistics of rainfall rate at a location of interest.
Research reported in (Yang, 2016) has produced a single general empirical equation that fits both the space correlation and the
time correlation functions. The common function is given by:
$\rho(x) = \frac{a}{a + x^n}$          (2)
where $x$ can either be $d$ which represents the distance in $km$ or $t$ which is the time lag in $mins$.
An empirical equation has been proposed in (Yang, 2011) that can give an excellent estimate of the probability of rain
occurrence ($P_0$) throughout the whole range of integration length. The mathematical equation is described by:
$P_0(x) = 100 - b\exp(cx^e)$          (3)
where $b$, $c$ and $e$ are experimental constants which can be determined from study and $x$ denotes either spatial integration length $L$
or temporal integration length $T$.
**3.2. Data Integration**
Following previous work (Yang, Oct 2011), the rainfall rate data can be up-scaled from short integration length to longer
one using:
$R_\lambda(x, y, t) = \frac{1}{\lambda^3 L^2 T} \int \int \int R(x', y', t') I \left( \frac{x - x'}{\lambda L}, \frac{y - y'}{\lambda L}, \frac{t - t'}{\lambda T} \right) dx' dy' dt'$      (4)

where $R_\lambda(x, y, t)$ is the rain rate at position $(x, y)$ derived from a spatial integration region of linear size $\lambda L$ and temporal
integration time $\lambda T$. $\lambda > 1$ is known as the scale parameter. More generally, the spatial and temporal regions could have
different scale parameters e.g.:
$R_\lambda(x, y, t) = \frac{1}{\lambda^2 \varphi L^2 T} \int \int \int R(x', y', t') I \left( \frac{x - x'}{\lambda L}, \frac{y - y'}{\lambda L}, \frac{t - t'}{\varphi T} \right) dx' dy' dt'$      (5)





The radar-derived rain rate data can be upscaled to coarser resolution based on above equations. It is important to highlight that
each grid point will be used only once for each integration and no overlapping regions are considered. The integrated data will be
tiled up without changing the size of original rain map but new dataset with larger integration scale will be achieved. Note that
the larger the integration length the smaller number of data samples will be. Particularly, it requires $\lambda$ and $\varphi$ must be integer to
enable this procedure. Therefore, it is notable that the integration length of the new data is the integral times of original radar
data, and it will be $\lambda L$ and $\varphi T$, here $L = 5\ km$ and $T = 15\ mins$.
**3.3. Approach for the Implementation of 3D Interpolation**
According to our previous work (Yang, 2011), we found that the rain characteristics regularly changing with increasing
integration length both in space and time domains. This interesting finding indicates that the studied rain characteristics at other
spatial or temporal integration lengths can be reasonably predicted using such regularity. More speculatively, it enables the 3D
interpolation to be achievable if there are enough measurements with different space-time resolution combinations.
Fig.2 shows the grid of available rain data points from the NIMROD radar measurements. The dots represent the available space-
time integration length combination where the rain characteristics can be computed based on available NIMROD data. The lines
represent the range of integration length where rain characteristics can be calculated from equation (1) to (3). It is notable that the
proposed statistical model in our previous work can only produce estimation of rain characteristics along the line but not the
blank area. Taking advantage of the regular distribution of the measurements, the key rain characteristics at other spatial-
temporal integration lengths can be predicted using any existing interpolation technique.

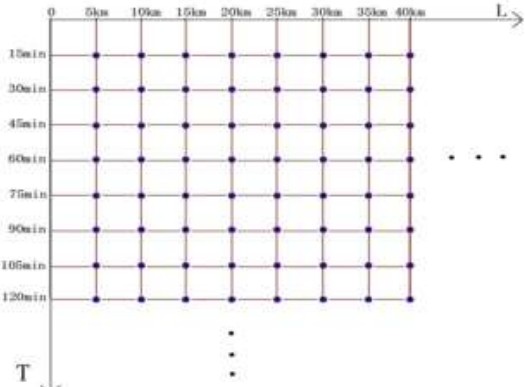


**Figure 2**: Grid of available rain data points from the NIMROD radar measurements. The dots represent the available space-time
integration length combination where the rain characteristics can be computed based on available NIMROD data.
**4. Experimental Results**
**4.1. Contour Map of Rain Characteristics**
The proposed statistical model can provide estimates of key rain characteristics (including the first order statistics of rain, the
spatial and temporal correlation of rain rate, as well as the probability of rain/no rain) in two dimensions. Considerable
computation is required to extract these summarizing statistics from the NIMROD databases. Based on the proposed model,
however, the rain characteristics at any data available locations within the Western Europe can be achieved. The work in this
paper has produced a multi-resolution database of parameters and contour maps that cover the whole of Western Europe. With
the help of this database, the user can easily obtain the characteristics of rain (or the distribution coefficients) at any location
within the studied area.





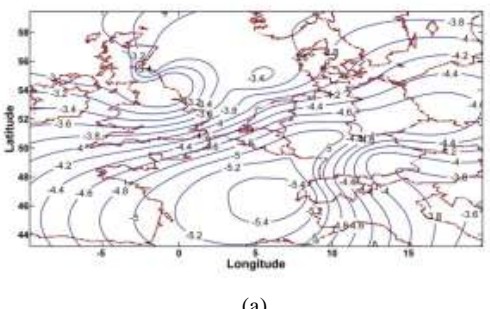 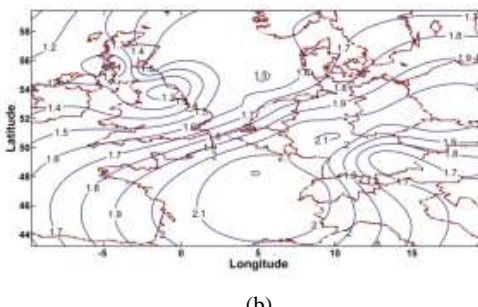


(a)                          (b)

**Figure 3:** Contour maps of rain distribution coefficients with spatial integration length of $5\ km$ and temporal integration length
$15\ min$: (a) a plot of $\mu$ values and (b) a plot of $\sigma$ values.
Example contour maps of the log-Normal rain rate distribution parameters $\{\mu, \sigma\}$ are presented in Fig. 3. Here, the longitude and
latitude values are achieved by using the approach given in Appendix A. Fig. 3(a) is the map of $\mu$ values cross the Western
Europe and Fig. 3(b) is $\sigma$ values. The background is the map of the Western European coastline, and the calculated parameter at
each individual location (here the spatial integration length is $5\ km$ and the temporal integration length is $15\ mins$) is
superimposed on the map. It shows that the contour map can provide the parameter value at any location within the range from
$-9.7370°$ to $19.8364°$ in longitude and between $43.1938°$ to $59.4306°$ in latitude. The calculated values are stored in a database
from which $\{\mu, \sigma\}$ can be easily obtained by simply inputting the longitude and latitude information for any desired location. This
is very convenient as almost no computation time is needed. Similar results of other rain characteristics have also been produced
and stored in the database, but not presented in this paper. In addition, the rain characteristics at other integration length
combinations between $\{5\ km, 15\ mins\}$ and $\{75\ km, 120\ mins\}$ have been computed and stored in the database. Given this
database, the prediction of the rain characteristics at some finer space-time resolutions can be estimated by interpolation.
**4.2. Prediction of Rain Characteristics in Space-Time**
The existing NIMROD radar maps have been integrated to some integration length combinations from $\{5\ km, 15\ mins\}$ to
$\{75\ km, 120\ mins\}$. The key characteristics of rain were then analysed to see how they vary with integration length. Table 1
gives an example of the probability of rain $(P_0)$ with a range of integration length combinations, at Portsmouth (UK).

**Table 1:** Probability of rain occurrence for increasing spatial-temporal integration lengths ranging from $5\ km$ to $75\ km$ and
$15\ mins$ to $120\ mins$ at Portsmouth.

| | $T$ | | | | | | | |
|---|---|---|---|---|---|---|---|---|
| $L$ | $15\ mins$ | $30\ mins$ | $45\ mins$ | $60\ mins$ | $75\ mins$ | $90\ mins$ | $105\ mins$ | $120\ mins$ |
| $5\ km$ | 15.0 | 22.9 | 25.9 | 28.5 | 30.5 | 32.6 | 34.3 | 36.1 |
| $10\ km$ | 23.4 | 28.61 | 32.3 | 35.3 | 37.7 | 40.1 | 41.9 | 43.8 |
| $15\ km$ | 28.6 | 33.7 | 37.5 | 40.3 | 42.9 | 44.9 | 46.7 | 48.6 |
| $20\ km$ | 32.6 | 37.9 | 41.6 | 44.7 | 47.1 | 49.4 | 51.4 | 53.1 |
| $25\ km$ | 35.2 | 40.6 | 44.1 | 47.2 | 49.5 | 51.7 | 53.7 | 55.2 |
| $35\ km$ | 42.5 | 48.3 | 52.2 | 55.4 | 57.9 | 60.1 | 61.2 | 63.1 |
| $40\ km$ | 45.9 | 49.4 | 53.2 | 56.1 | 59.6 | 62.6 | 64.4 | 67.9 |
| $45\ km$ | 48.8 | 55.6 | 59.5 | 62.2 | 64.7 | 66.6 | 68.2 | 69.6 |
| $50\ km$ | 49.9 | 56.2 | 60.2 | 62.9 | 65.6 | 67.2 | 69.5 | 71.8 |
| $55\ km$ | 56.5 | 62.2 | 65.8 | 68.5 | 70.7 | 72.2 | 73.5 | 75.3 |
| $65\ km$ | 60.9 | 65.5 | 68.5 | 70.6 | 72.3 | 74.2 | 75.1 | 76.5 |
| $75\ km$ | 63.9 | 69.1 | 72.4 | 74.4 | 76.3 | 77.8 | 78.7 | 79.9 |





It shows that the $P_0$ value changes with increasing spatial-temporal integration length. Similar results can be found for other
studied parameters. These data allow the prediction of parameters at other space-time resolutions. The top-left hand corner of the
table is the computed value with the shortest available spatial-temporal integration length ($\{5\ km, 15\ mins\}$) derived from EU
NIMROD radar, and the right-hand bottom corner is the coarsest one ($\{75\ km, 120\ mins\}$) after integration. From Table 1, one
can see that the characteristics of rain change systematically with increasing integration length. Given this finding the predictions
at finer resolution can be estimated by interpolation.
In this study, the cubic spline interpolation algorithm has been chosen to implement this task. The cubic spline is a function that
is constructed by piecing together cubic polynomial on different intervals (Keys, 1981). It has the form
$$S(x) = \begin{cases} s_1(x) & if\ \ x_1 \le x < x_2 \\ s_2(x) & if\ \ x_2 \le x < x_3 \\ \quad ... \\ s_{n-1}(x) & if\ \ x_{n-1} \le x < x_n \end{cases} \tag{6}$$
where $s_i$ is a third degree polynomial defined by:
$$s_i(x) = a_i(x - x_i)^3 + b_i(x - x_i)^2 + c_i(x - x_i) + d_i \tag{7}$$

Cubic spline is often used for 1D interpolation. The data in each row and column of the database (see the example in Table 1)
can be treated as samples in one dimension. It enables the use of cubic spline interpolation to estimate parameter values at other
scales, based on the measured parameters. The first step is to extract the multi-scale parameters for a desired location from the
database. Cubic spline interpolation is then used to interpolate to a different spatial or temporal integration sizes. In this study,
the "bicubic" interpolation algorithm in MATLAB was used. Mathematically, the bicubic interpolation, which is an extension of
1D cubic interpolation, is used to interpolate data points on a two dimensional regular grid. It can be accomplished using cubic
spline algorithm (we provide part of the software program in Appendix B to show the approach of 3D space-time interpolation).
The software proposed in this work uses the produced parameters' database. It contains the fitted rain parameters for a range of
integration lengths between $\{5\ km, 15\ mins\}$ and $\{75\ km, 120\ mins\}$ for the whole of the studied area (Western Europe). The
software extracts the rain characteristics with all available integration lengths at the location of interest. Taking the extracted data
as input values, the interpolation algorithm then processes the data and gives the prediction at other space-time resolutions. Note
that this is true only for the locations for which radar measurements data is available (the black area in Fig. 1).

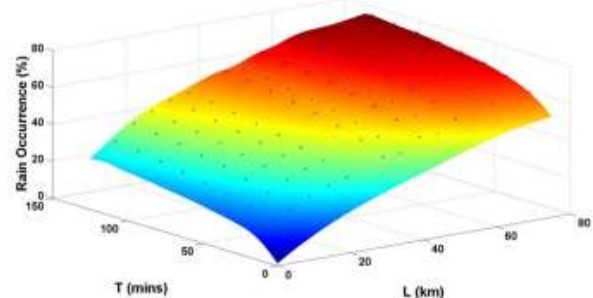


**Figure 4:** An example of $3D$ space-time interpolation of $P_0$ at Portsmouth.
Fig. 4 presents the example of predicted probability of rain occurrence at other spatial-temporal integration lengths, along with
the measured data in Table 1, for Portsmouth. It is clear that the outcome of the $3D$ interpolation is a surface constructed from
many $2D$ curves both in space and time domains. The dots are the measured values at a range of spatial-temporal integration
lengths that are multiples of the data resolution, whilst the surface is produced by the interpolation algorithms to be consistent



with these data. The multi-scale data are regularly spaced, which reduces the complexity of the interpolation algorithm.
Interestingly, the results show that $P_0$ values increase systematically with increasing spatial-temporal integration length. In
addition, by interpolation the values at resolutions smaller than $\{5\ km, 15\ mins\}$ can also be predicted. The exterpolation can be
constrained by the assumption that $P_0 \to 0$ as either $\lambda \to 0$ or $\varphi \to 0$. This enables the predictions to be plotted smoothly to form
a $3D$ surface. The resolution of the studied key characteristics of rain offers significant improvements over previous methods
(e.g. Bell, 1987) and it is these that are important for rainfall field simulation studies in future. The salient point of the proposed
interpolation approach is that the best estimate can be obtained with high accuracy for the space and time resolutions up to $50m$
and $6s$, respectively. Predictions finer than this threshold are unacceptable as negative data is produced. This is impossible due to
the $P_0$ should not less than 0. Other interpolation technique might give better results but this is not covered in this paper. The
validity of the interpolated parameters needs to be tested, and this is limited by the availability of data at small spatial and
temporal integration volumes. One test that can be performed is to use $\{5\ km, 15\ mins\}$ EU NIMROD data to predict the
distribution and correlation functions of $\{1\ km, 5\ mins\}$ UK NIMROD data.
**5.    Validation**
The absence of measured data at the smaller space-time scales causes great difficulties in validating the proposed method.
However, the $\{1\ km, 5\ mins\}$ UK NIMROD radar measurements can be used to address this issue to some extent. In this paper,
the key rain characteristics at Portsmouth have been estimated at scales of $\{1\ km, 5\ mins\}$  and these were compared with
interpolations from the EU NIMROD data.

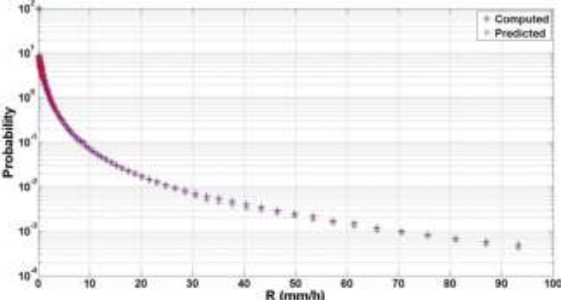


**Figure 5:** A comparison exceedance distribution of rainfall rate estimated by interpolation from $5\ km$ data to $1\ km$ data and
estimated directly for $1\ km$ data.
Fig. 5-6 present comparisons of rainfall rate characteristics estimated by extrapolation from $5\ km$ EU NIMROD to $1\ km$ data
and estimated directly from $1\ km$ UK NIMROD data. The predicted $\{\mu, \sigma\}$ values are $\{-3.75, 2.12\}$ and the computed values
are $\{-3.85, 2.04\}$. The predicted probability of rain occurrence ($P_0$) and measured one are 12.5% and 12.1%, respectively.
Although the $\{\mu, \sigma, P_0\}$ values of both are marginally different (2.7%, 3.8%, 2.7% differences for $\mu, \sigma, P_0$, respectively), the
associated 0.1%, 0.01% and 0.001% exceeded rain rates are similar, this can be seen in Fig. 5. In particular, the proposed model
gives excellent approximation for the first-order rainfall rate statistics, especially for the rain rate lower than $40\ mm/h$ for which
the accuracy is higher than 90%. The probability of heavy rain event is extremely low so that there is no sufficient data is
available. This results in the higher bias for the range where  $R > 40\ mm/h$. Fig. 6(a) shows that the spatial correlation using the
predicted values is in agreement with the computed values. There is a small difference between the temporal correlation
functions of rain rate using predicted data and measured data at short time lags up to roughly $150\ mins$, see Fig. 6(b). However,
the result is still acceptable as the trend is similar, especially for large time lags. This shows that the approach proposed in this



paper has potential and requires considerable less computational effort than the direct estimation of these distributions from the
data. However, the rain characteristics at scales finer than $\{1\ km, 5\ mins\}$ cannot be validated due to lack of radar/raingauge
data.

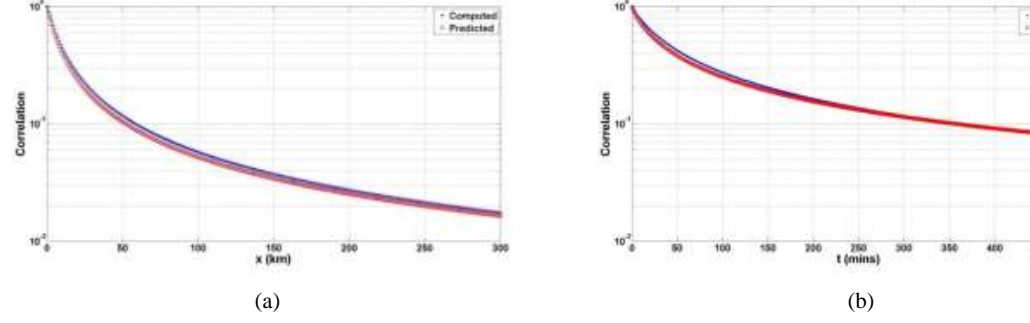


(a)                                                                (b)

**Figure 6:** A comparison of correlation function of rainfall rate estimated by interpolation from $5\ km$ data to $1\ km$ data and
estimated directly for $1\ km$ data: a) spatial correlation function; b) temporal correlation function.

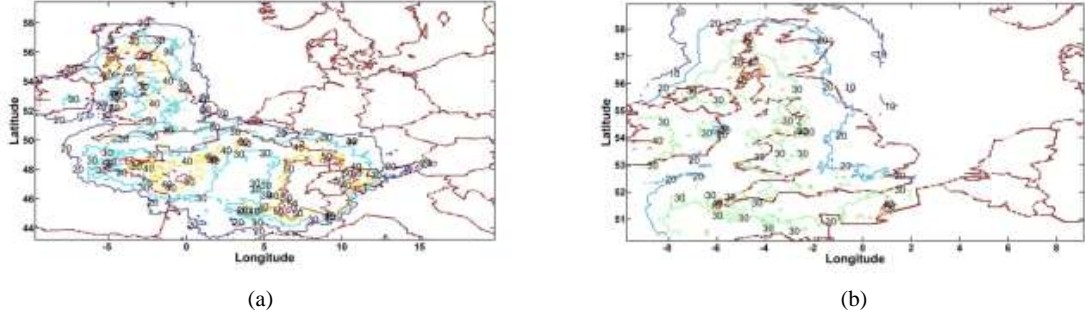


(a)                                                                (b)

**Figure 7:** Contour map of 0.01% exceeded rain rates: a) predicted by interpolation from $5\ km$ EU NIMROD to $1\ km$; b)
measured from $1\ km$ UK NIMROD.

Fig. 7(a) shows the map of 0.01% exceeded rain rates across the Western Europe predicted by interpolation from the $5\ km$ EU
NIMROD to $1\ km$. The results are plausible for most areas. However, it shows that radars accuracy is affected in the Grand
Massive alpine area of France. Fig. 7(b) presents the map of 0.01% exceeded rain rates across the British Isles given by the
$1\ km$ UK NIROMD. Note that the rain rate with 0.01% exceedance in both figures tends to reduce towards the edge of the radar
region and this is almost certainly an artefact. It could be due to how the contour function deals with NaN (caused by data
unavailable); or something to do with the data at the edge of the radar network. The contour map of 0.01% exceeded rain rate of
the average year given by ITU-R P 837-6 (ITU, 2013) is presented in Fig. 8. These two figures (Fig. 7 and Fig. 8), illustrate that
the results of the statistics in Fig. 7 are very similar. This indicates that the proposed model can give a reasonable estimation of
rain parameters that can be used to produce rain rates with 0.01% exceedance. However, the rain rate statistics given by ITU-R P
837-6 seems quite larger compared with the results from EU NIMROD data interpolate from $5\ km$ to $1\ km$ and estimated
directly for $1\ km$ data. This suggests that the ITU. Rec tends to over-estimates rain. Indeed, the overestimation of ITU-R P.837-
6 is likely also due to the overestimation of the rain amounts over oceans as obtained from the ERA-40 data produced by the
ECMWF (i.e. the input maps on which the ITU-R rain rate models relies on). This is why the ITU. Rec recommends users to use
their own data in order to produce better results.





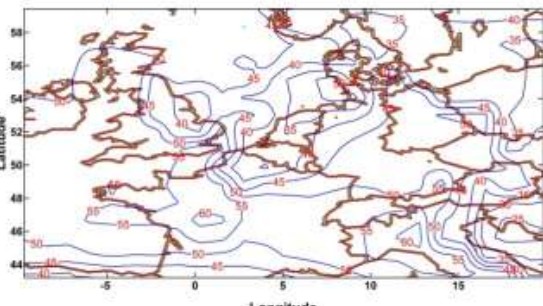


**Figure 8:** Contour map of 0.01% exceeded rain rates of the average year given by ITU-R P 837-6.
The differences between EU contour map, UK contour map and ITU contour map have been studied to show how accurate the
proposed approach is. Fig. 9 presents the contour map of the difference of rain rates with 0.01% exceedance based on the EU
data minus UK data. It shows that the proposed approach tends to overestimate the rain rates over land (see the example in
middle area of Fig. 7(b), but under-estimates over the ocean/sea areas (see the left-bottom area of Fig. 7(b)). However, the
difference is acceptable as it is in the range $2 - 5 \, mm/h$ for most areas. For some areas, the difference can up to $10 \, mm/h$ or
higher, but this is rare.

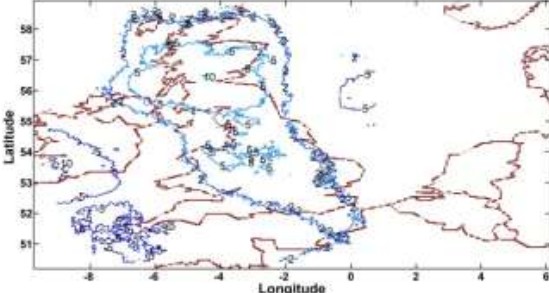


**Figure 9:** Contour map of 0.01% exceeded rain rates difference between the prediction from proposed approach and the
measurements from $1 \, km$ UK NIMROD.

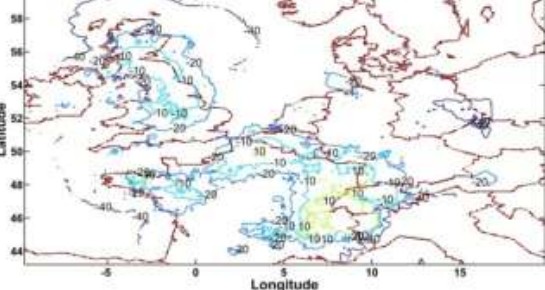


**Figure 10:** Contour map of 0.01% exceeded rain rates difference between the prediction from proposed approach and ITU-R P

837-6.

Fig. 10 presents the difference between the prediction from the proposed approach and ITU-R P 837-6 (EU predicted rain rates
minus ITU predicted rain rates). The contour map shows that the ITU-R P 837-6 tends to over-estimate rain rate compare to the
proposed approach for most areas. The difference can up to $40 \, mm/h$ for some regions. This indicates that the proposed



approach gives more plausible estimates than ITU-R P 837-6, although it is restricted to Western Europe. However, it is
necessary to highlight that for the Grand Massive alpine area of France, the proposed approach gives larger rain rates exceedance
than ITU-R P 837-6. This indicates that it is hard to give accurate rainfall rate measurements or prediction over mountain area
due to the difficulties associated with obtaining accurate rain radar readings (Johansson and Chen, 2003).
Fig. 9 and Fig. 10 present the visual comparison of 0.01% exceeded rain rates difference between the prediction from proposed
approach and the measurements from $1\ km$ UK NIMROD and ITU-R P 837-6. However, the error function can give more
information to the model performance validation. According to (Paulson et al., 2015 and ITU, 2013), the error function can be
defined as:
$$Error = \left| ln\left(\frac{R_{measured}}{R_{predicted}}\right)\right| \tag{8}$$
where $R_{measured}$ and $R_{predicted}$ are the measured and predicted rainfall rate with 0.01% exceedance, respectively. The error at
each individual location therefore can be calculated by Eq. (8).
Fig. 11 shows the error contour maps at 0.01% exceeded rain rate over the UK for both the proposed approach and ITU-Rec
model. Theoretically, the smaller the error value, the more accurate the model prediction will be. Fig. 11(a) shows that the error
of the proposed approach is between $0.02$ and $0.15$. It indicates that the approach proposed in this paper can produce reasonable
prediction. However, the error from ITU-R model can be up to nearly 2, see Fig. 11(b). Such high error value suggests that it is
better to use the local rain radar measurements for the model development if the data is available.
The mean error $\overline{Error}$ is calculated by:
$$\overline{Error} = \frac{1}{n}\sum_{i=1}^{n} Error_i \tag{9}$$
where $Error_i$ is the error for individual location and $n$ is the location index. The $\overline{Error}$ for ITU-R model is 0.62. It is roughly 7
times that the proposed approach for which the $\overline{Error}$ is as low as 0.09.

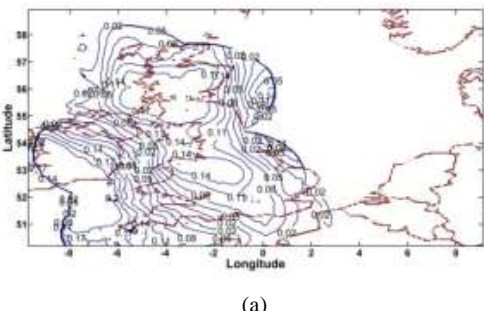 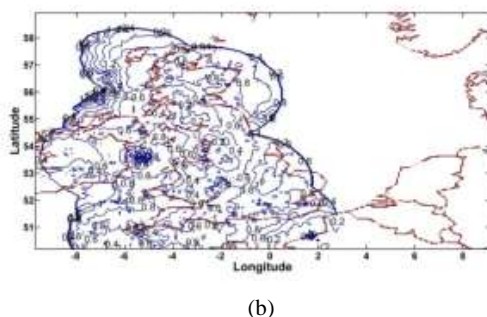

(a)                                                                     (b)

**Figure 11:** Contour map of error at 0.01% exceeded rain rate: (a) error distribution of proposed model, and (b) error distribution
of ITU-R P 837-6.

**Table 2:** RMSE of lognormal rain distribution parameters $\{\mu, \sigma\}$ in both space and time domains at four locations.

| Portsmouth |        | Paris  | Rennes | Reims  | Brussels | Zurich |
|------------|--------|--------|--------|--------|----------|--------|
| $\mu$      | 0.0761 | 0.0632 | 0.0945 | 0.0368 | 0.0526   | 0.0875 |
| $\sigma$   | 0.0485 | 0.0418 | 0.0323 | 0.0477 | 0.0392   | 0.0417 |


In particular, the Root-Mean-Squared Error (RMSE) has been applied to measure the goodness of fit between measured
lognormal parameter $\{\mu, \sigma\}$ obtained from radar-derived statistics and predicted values. The RMSE is defined as





$$E = \sqrt{\frac{\sum_1^n (P_v - M_v)^2}{n}}$$ (10)
where, $P_v$ and $M_v$ are predicted and measured values, respectively, and $n$ represents the number of samples. Table 2 gives the
calculated RMSE of lognormal rain distribution parameters $\{\mu, \sigma\}$ at six locations cover different climates within the studied
area. The small RMSE (less than 0.1) suggests that the proposed algorithm yields accurate predictions, especially for $\sigma$.

**6. Conclusion**
A simple but efficient interpolation/extrapolation approach has been presented. Instead of the radar-/raingauge-derived rainfall
rate data, the analyzed rain characteristics and fitted coefficients are used to predict rain at many space-time resolutions.
Databases with estimated parameter values, and maps for Europe, have been created to allow users to access the key rain
characteristics at any location within the study area. This provides great assistance to users as the rain characteristics can be
easily obtained without long computation. In particular, an approach to interpolate the fitted coefficients and/or rain
characteristics in space-time domain with arbitrary integration length has been proposed. Although parameters can be estimated
at any combination of spatial and temporal integration lengths by interpolation or/and extrapolation, the results have only been
tested down to $1\ km$ spatial. The predictions have been validated through comparing with the measurements from UK NIMROD
data. The results show that there is a reasonable agreement between the predicted and computed values. However, the predictions
with resolution finer than $\{1\ km, 5\ mins\}$ cannot be validated due to lack of radar/raingauge data.
Finally, the contour map of 0.01% exceeded rain rates cross Western Europe and the British Isles have been generated and
compared using the data interpolated from $5\ km$ to $1\ km$ and estimated directly from $1\ km$ data. The results are also compared
with ITU-R P 837-6 estimations.
**ACKNOWLEDGMENT**
The authors thank the British Atmospheric Data Centre (BADC), which is part of the NERC National Centre for Atmospheric
Science (NCAS), and the British Met Office for providing access to the NIMROD rain radar data sets (http://badc.nerc.ac.uk/).
Partial support from ICT COST action IC0802, "Propagation tools and data for integrated telecommunication, Navigation and
earth observation systems" is gratefully acknowledged.

**Appendix A: Calibration of NIMROD data**
The calibration of NIMROD data is significant for this study. By choosing some samples (normally the more samples that are
chosen the more accurate the result will be, here the author use 30 samples), two algebraic equations are used, one is for latitude
and the other one is for the longitude. These two numerical equations could allocate the roughly latitude and longitude values for
different locations of Western Europe.
For the development of the relative algebraic equations, the general procedures are summarised as following steps:
1. Choosing some radar images from NIMROD data set
The NIMROD radar-derived rain maps are helpful and critical for the calibration therefore some maps should be selected at the
initial stage. The maps need to meet the following requirements.
i) There is not too much rain in the selected map, the less the better. Under this circumstance, it could be easier to find some
small rainy areas or even single rain point (ideal situation) from the map. In this way the error can be greatly reduced.
ii) The separation of different rainy areas in the same map should be large enough; otherwise, it is easy to make a mistake when
trying to find out the corresponding coordinate (row and column) in the grid.





2. Allocating the selected samples
This piece of work used a map of the Europe (not the NIMROD radar map) that has accurate latitude and longitude information.
As to the scale, ideally, is the finer the better. Based on this an accurate result can be achieved. In this study, the finest precision
of the European map used to provide the latitude and longitude information is $20\ mins$. Through comparing the radar images
and the used map, the locations of the selected samples can be physically allocated on the map. In addition, both the latitude and
longitude values of the location of interest can be read and recorded as it is visible on the map.
The achieved latitude and longitude values of all selected locations can be transformed into degrees by using the following
mathematical equation:
$Finalvalue = X + \frac{Y}{60} + \frac{Z}{3600}$ (A1)
3. Fitting the line
It is difficult to get the real latitude and longitude value for the location of interest since error is unavoidable. However, by
making use of the achieved data from the selected samples, a reasonable line to offset and reduce the error can be proposed.
The final equations are given as follows:
$y_{(longitude)} = 0.0658x - 19.8364$ (A2)
$y_{(latitude)} = -0.0409x + 59.430$ (A3)
Here $x$ denotes either row or column number of the NIMROD data grid with spatial integration length of $5\ km$, and $y$ is the
corresponding coordinate value in either latitude or longitude. The fitted lines are shown in Fig. A.

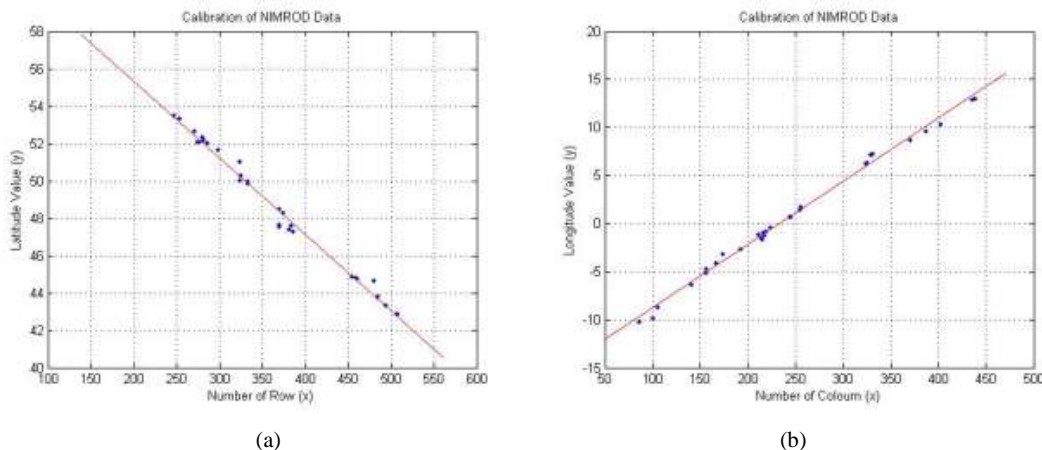


(a)                                                                    (b)

**Figure A**: (a) tendency of latitude changing with distance, (b) tendency of longitude changing with distance.

Viewed from the software generated figure (see Fig. A), it is clear that the fitted lines are straight. Fig. A(a) shows that the slope
for the latitude is negative. The reason is that the origin of the data matrix for rain field image is starts from the top left to bottom
right. It means that the smaller the row number (the value of $x$), the higher the latitude value. In other words, the latitude value
decreases with the increasing row number. Fig. A(b) shows that the slope for the longitude is positive. Noticeably, the larger the
column number (the value of $x$), the higher the longitude value. Here, it is important to highlight that the longitude values can be
either positive or negative. The reason is that the Prime Meridian goes across the studied map.

**Appendix B: Code for extrapolating the measured data**





```
function Zi=SpaceTime_Interpolation(Data,ZI)
  GG = zeros(length(Data),length(Data));
for i = 1 : length(Data)
    for j = 1 : length(Data)
        if i ~= j
            magx = sqrt((Lat(i)-Lat(j))^2 + (Lon(i)-Lon(j))^2);
            if magx >= 1e-7
                GG(i,j) = (magx^2) * (log(magx)-1);
            end
        end
    end
end
m = GG\Z; % Compute model "m" where data "d" is equal to "Z"
%    Find 2D interpolated surface through irregular/regular grid points
gg = zeros(size(m));
for i = 1 : size(ZI,1)
    for j = 1 : size(ZI,2)
        for k = 1 : length(Data)
            magx = sqrt((YI(i,j)-Lat(k))^2 + (XI(i,j)-Lon(k))^2);
            if magx >= 1e-7
                gg(k) = (magx^2) * (log(magx)-1);
            end
        end
        ZI(i,j) = sum(gg.*m);
    end
end

for n=1:30
    ZI(Lat(n),Lon(n))=Sample(n);
end
    %   Plot result if running example or if no output arguments are found
mesh(YI,XI,ZI)
Zi=ZI; % Zi includes all the extroplated data that forms the 3D surface
```

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
