# Peer review of "A New Approach for Rainfall Rate Field Space-Time Interpolation for"

_Hydrology and Earth System Sciences, 2018_

## Referee Comment (RC1) · G. Pegram (Referee) · 24 Aug 2018

Yang: hess-2018-343

This paper contains passages from the lead author's PhD thesis submitted in 2016, in which there are references to 3 of his articles, co-authored with his supervisors. One article was a conference abstract and after a web-search it appears that the other two have not been accepted for publication by the IEEE, Radio Science, nor any other journal. I read the thesis through so I could be properly informed and found that the content of this article has been sourced from it almost without alteration. The figures in the article are the same as the thesis, but have been copied several times and are blurred and illegible. In addition, it is important to note that the thesis was aimed at

the field of signal processing, and in my opinion does not have a home in hydrology. My reasons are that maps of parameters of the censored lognormal distribution, fitted to a range of spatial and temporal intervals, but taking no notice of seasonality, may be interesting for electronic engineers checking attenuation of transmission veracity, but they are presumably looking for average, not particular temporal behaviour as is needed in hydrological work.

I recommend rejection.

Geoff Pegram 24 August 2018

———————————————————

---

## Editor Comment (EC1) · U. Ehret (Editor) · 11 Oct 2018

Dear Authors, dear Referee,

I talked to the Editor-in-Chief Erwin Zehe about the referee reporting large overlap (content and wording) of the submitted manuscript and the first author's PhD thesis published at the University of Portsmouth. The conclusion is: All EGU journals have the policy that while plagiarism and self-plagiarism are invalid in general, this does not apply to PhD theses that are so far only published as grey literature (e.g. in an institute's series of publications). This applies to the manuscript submitted by the authors. I will therefore not reject the manuscript for plagiarism, but will instead decide based on the referee's evaluation of the content.

[Figure]

Yours sincerely,

Uwe Ehret
* * *

---

## Referee Comment (RC2) · Anonymous Referee #2 · 15 Oct 2018

This paper is a re-writing of a recent thesis from the area of high frequency satellite communication networks and their dependence on precipitation. The goal is to provide high-res space-time fields of precipitation, using a simple but efficient interpolation/extrapolation approach. From what I have understood, and that is not much, the approach interpolates, using cubic splines, basic spatio-temporal statistics to any given degree, in order "to predict rain at many space-time resolutions".

The original thesis' topic, its context, referencing and methodology, is foreign to hydrology, which would have required a thorough and thoughtful adaptation of concepts and procedures by the authors. This was not done, however, nor could I identify much effort to do so. Although I am sure that there are many parallels in the core concepts about precipitation, a reviewer from outside the field of satellite communications will

have a very hard time of doing that translation all by him/herself. And so will the reader of HESS. This applies even more since the main concepts and equations are incompletely presented, with a number of missing definitions, as outlined partly below.

I may be completely missing the point, but the overall approach of interpolating basic long-term statistics to higher resolution is everything but new, but it is a totally different task of doing a similar thing for the single rainfall events, which would probably be what is required by satellite communication but which was not provided here.

Anyway, my main point is that the paper is not written in a way that is comprehensible to the average hydrologist, and by the same reason, unreviewable by me. All I can do is recommend rejection for publication in HESS. There is a chance, though, that after a complete re-write in the form of a truly hydrologic paper, which includes a thorough review of the relevant hydrological literature and a consistent and complete mathematical exposition of its core concepts, the study may eventually be publishable.

Some detailed comments:

11: why satellite network systems?

25: this is really but one of many distributions, I would guess at least a dozen.

32: for this motivation to be understandable by a larger audience, especially of hydrologist and other natural scientists, it is required to explain terms such as integration volume and length in more hydrological or meteorological terms.

35: I wonder what a hydrologist feels like when she or he sees that the original thesis has been copied with no adjustments at all here.

43: I don't think that this is an interpolation technique.

49: what is space-time averaging theory?

49: the meaning of 'downscaling' as used here differs from other uses based on climate downscaling; this should be clarified.

55: what is a 'static nonlinear transformation'?

59: again, this is NOT an interpolation but a disaggregation.

61: "can easily capture any moment of the observed signal" – statements like this are almost certainly not true.

69: what is EM wave propagation?

89: How will you preserve statistical properties that are unobserved?

124. I cannot find any such discussion, nor section 0.

126. Like above, overly general statements like this one are barely verifiable and should be avoided.

128: What is $Q_{inv}$?

132: with enough tolerance one can fit anything, so again, how verifiable is this? - Is 'a' and 'n' fitted separately? By choosing the right units (such as km and min) one can probably bring them into a match.

134: 'excellent': see comments above.

136: what probability is decsribed here? The conditional probability for rain at distance x GIVEN it rains at x=0? - And applies this to any rainfall amount?

140: I could not find the referenced work in Yang, 2011.

142: what is I?

179: 1-1 copy from thesis

197: 1-1 copy from thesis